# A Review of the Paleobiology of Some Neogene Sharks and the Fossil Records of Extant Shark Species

**Olaf Höltke \*, Erin E. Maxwell** 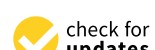 **and Michael W. Rasser**

Staatliches Museum für Naturkunde Stuttgart, Rosenstein 1, 70191 Stuttgart, Germany;
erin.maxwell@smns-bw.de (E.E.M.); michael.rasser@smns-bw.de (M.W.R.)
\* Correspondence: olaf.hoeltke@smns-bw.de

**Abstract:** In recent years, new findings and new methods (stable isotopes of oxygen, zinc, and nitrogen; 2D and 3D modeling; and geometric morphometric analyses of the teeth) have enhanced our knowledge of the Neogene shark fauna and its paleobiology. Several papers deal with the large *Otodus* (*Megaselachus*) species, including the construction of a 3D model, as well as insights into its lifestyle and diet. In addition, the skeletal remains of *Carcharias gustrowensis*, *Carcharodon hastalis*, and *Keasius parvus* and a natural tooth set of *Carcharodon hubbelli* have been described in the last 13 years, and the dentition of the Neogene species *Carcharoides catticus*, *Megachasma applegatei*, and *Parotodus benedenii* has been reconstructed. Stable isotope analyses of the teeth from the Neogene species of *Araloselachus*, *Carcharias*, *Carcharodon*, *Galeocerdo*, *Hemipristis*, and *Mitsukurina* have given insights into the trophic positions of these genera during the Neogene, and shark teeth preserved near the skeletal remains of prey animals (mammals) and shark bite traces on these remains provide direct evidence of trophic interactions. The tooth shape, fossil locality, and paleoenvironment have been used to better understand the taxa *Carcharhinus dicelmai*, *Megalolamna paradoxodon*, *Pachyscyllium dachiardii*, and *P. distans*. Among extant species, *Galeorhinus galeus* can be traced back to the Eocene. *Alopias superciliosus*, *Rhincodon typus,* and possibly *A. vulpinus* can be traced back to the Oligocene. Species present by the Miocene include *Alopias vulpinus*, *Carcharhinus amblyrhynchoides*, *C. amblyrhynchos*, *C. albimarginatus*, *C. amboinensis*, *C. brachyurus*, *C. brevipinna*, *C. falciformis*, *C. glaucus*, *C. leucas*, *C. limbatus*, *C. longimanus*, *C. macloti*, *C. obscurus*, *C. perezi*, *C. sealei*, *Centrophorus granulosus*, *Cetorhinus maximus*, *Dalatias licha*, *Deania calcea*, *Galeocerdo cuvier*, *Glyphis glyphis*, *Heptranchias perlo*, *Isurus paucus*, *Lamna nasus*, *Negaprion brevirostris*, *Odontaspis ferox*, *Pseudocarcharias kamoharai*, *Sphyrna media*, *S. mokarran*, and possibly *Carcharodon carcharias*. First appearing in the Pliocene are *Scymnodon ringens*, *Somniosus rostratus*, and *Zameus squamulosus*. For some extant species (*Carcharias taurus*, *Hexanchus griseus*, *Isurus oxyrinchus*, *Notorynchus cepedianus*, and *Sphyrna zygaena*), it is not clear whether the assigned Neogene teeth represent the same species. The application of new methods to more fossil shark taxa, a detailed search for shark fossils, and better knowledge of the dentition of extant species (especially those with minute-sized teeth) will further enhance our knowledge of the evolution and paleobiology of sharks.

**Keywords:** Selachii; Miocene; Pliocene; paleobiology; ecology; Recent; *megalodon*

## 1. Introduction

The earliest record of elasmobranch fishes is probably from isolated scales potentially referable to the chondrichthyans, which date back to the Late Ordovician Epoch, about 455 million years ago [1]. Apart from a different tooth shape, Paleozoic sharks had a different anatomy from "modern" sharks (Neoselachii), which are known from the beginning of the Mesozoic era. Four key differences separating neoselacians from Paleozoic sharks were mentioned [2,3]: The jaws of neoselachians open wider than in earlier forms because of the greater mobility in the jaw joint and a highly kinetic palatoquadrate and hyomandibular. The notochord is enclosed in and constricted by calcified cartilaginous

vertebrae, whereas primitive chondrichthyans had a simple notochordal sheath. The limb girdles in neoselachians are strengthened by fusion or firm connection on the midline, which allows for more powerful muscle activity. The basal elements (the radials) in the paired fins are reduced, and most of the fin is supported by flexible collagenous rods called ceratotrichia or actinotrichia. However, the general phylogeny, synapomorphies, evolution, and origin of elasmobranchs (sharks and rays) and holocephalans are still under discussion [4–7]. The rise and diversification of Neoselachii began in the Early Triassic Epoch, and, by the Neogene period, the shark fauna was similar to the Recent fauna. However, despite general similarities, the timing of the appearance of extant morphospecies, the extinction of some Paleogene–Neogene species, and the potential trophic changes resulting from these origin and extinction dynamics can provide insights into the structure and occupancy of higher trophic levels in Recent oceans.

The cartilaginous skeleton of sharks is normally not preserved in the fossil record, making the teeth the most abundant records of fossil sharks. Sharks replace their teeth continuously throughout their lifetime, and this high production of potential bioclasts makes fossil shark teeth the main vertebrate fossils in marine deposits of the Paleogene and Neogene periods. Therefore, the designation of species is mostly based on a few isolated teeth. In some cases, calcified vertebral centra can be found, as well as dermal denticles, fin spines, and gill rakers. The skeleton, or parts of it, were only fossilized under specific environmental conditions (e.g., fast sedimentation and exclusion of oxygen). Accordingly, such findings are very rare [8–11].

The "classical" method to infer the shark ecology from teeth is to look to extant relatives as analogues, as well as the shape of the teeth. The teeth were divided the different tooth shapes into eight adaptive dental types [3]. In addition to the tooth size and shape, the embedding sediment also gives an indication of the habitat preferences of Neogene sharks. In the last 20 to 30 years, new findings, as well as new methods, have made it possible to obtain more detailed information on the paleoecology of Neogene sharks. Recently, It was quantified the classical method by applying 2D geometric morphometrics to statistically discriminate the diet based on tooth shape, and it was also determined that variations in tooth morphology could be partitioned into seven key variables with which ecological roles in fossil sharks could be accurately assessed [12,13]. Paleobiology is probably best documented for the most famous fossil shark, *Otodus* (*Megaselachus*) *megalodon*, simply because there have been so many recent papers with this species as the main subject. The aim of this paper is to provide a detailed overview of those Neogene shark species for which the most data are available, excluding taxa described from only one or a few teeth. We then summarize what is known of the paleobiology of these Neogene shark species, as well as examining the first appearance of Recent species in the Neogene period (or sometimes earlier). The classification is based on Cappetta [3]. Genera and species are arranged in alphabetical order within higher taxonomic groupings. Lastly, we provide an outlook on possible future developments concerning research on fossil sharks. This work presents the current state of the art concerning the paleobiology of Neogene sharks, as well as the fossil records of extant species.

## 2. Methods Used to Infer the Paleobiology of Fossil Sharks

There are six methods commonly employed to reconstruct the paleobiology of fossil sharks.

1. The "classical" method of inferring the diet based on the teeth, as mentioned above. More discoveries have made it possible to reconstruct complete dentitions and infer the diet with greater accuracy. Complete dentitions, also called tooth sets, are a more solid framework with which to reconstruct the diets of the sharks than isolated teeth [14]. There are three types of tooth sets [14]. (a) In a natural tooth set, the jaw is preserved, and all of the teeth are in their original positions. This the best but also the rarest condition. (b) An associated tooth set is one based on the teeth of an individual shark, where the teeth are found displaced from their natural positions. This is also

rare and mostly associated with skeletal remains [10]. (c) An artificial tooth set can be constructed from a number of tooth types from one locality that are believed to belong to one species. The teeth probably come from different individuals. This is the main type of reconstruction.

2. The rare discovery of preserved articulated or disarticulated skeletons or parts thereof, including body proportions, gastric contents, and data on reproductive biology [11].

3. Bite marks on fossil bones or shark teeth embedded next to the fossilized skeletal remains of prey animals can also be used to provide direct evidence of predation or scavenging [15,16].

4. Stable isotopes can be used to reconstruct trophic positions [17,18].

5. Two-dimensional or three-dimensional computer modeling based on vertebral centra and morphometric comparisons with Recent sharks can provide information on body size and tooth shape [19,20].

6. The shape and morphology of the placoid scales can be used to reconstruct swimming abilities [21].

## 3. Materials and Methods

For this review, the literature was searched for information concerning the ecology and paleobiology of extinct Neogene shark species, as well as for the referral of fossil remains to extant species [22]. Although the focus of this paper is on Neogene shark species, when the first occurrence of extant species predates the Neogene, this is nevertheless also noted. Throughout this review, extinct shark species are labeled with a dagger symbol (†) for clarity. Geologic ages can be found in Figure 1.

| Pleistocene | | |
|---|---|---|
| Plioc. | Late | Piacenzian |
| | Early | Zanclean |
| Miocene | Late | Messinian |
| | | Tortonian |
| | Middle | Serravallian |
| | | Langhain |
| | Early | Burdigalian |
| | | Aquitanian |
| Oligocene | | |
| Eocene | | |

**Figure 1.** Stratigraphic table.

The fossil record of Recent species is documented. In addition, when remarkable information concerning the biology of Recent species has been discovered from fossil sources, e.g., a dietary shift, this is mentioned in the text. Otherwise, the reader is referred to the according literature, because details of the ecology of extant sharks are well documented

elsewhere. Likewise, extant species are not considered because photos of them can be found in nearly every scientific or non-scientific book on sharks.

For each Neogene shark species, one fossil tooth has been illustrated, or, in the case of the extinct basking shark *Keasius parvus*, a gill raker (Figures 2 and 3). The latter species first appears in the Oligocene (Paleogene) and the identified raker is from this epoch simply because it was the best-preserved one available to the authors. However, a complete preserved tooth was not available for every taxon. The extinct Neogene shark species and the according methods used to infer their paleobiology are summarized in Table 1. Many of the teeth of the extinct Neogene shark species mentioned in this paper have a nearly global distribution. It was therefore decided not to list all of their fossil discovery localities, as was done for the fossil records of extant sharks, in order to constrain the length of this paper. Despite the large volume of research on fossil sharks undertaken during the past few decades, there are unresolved questions and different opinions, especially concerning the genus-level membership of some taxa. However, a discussion of the problems regarding Neogene taxa is beyond the scope of this paper, and it is not relevant to this review. Details of these debates can be found in the cited literature.

**Table 1.** Extinct (†) Neogene shark species and the methods used to infer their paleobiology.

| Extinct Neogene Shark Species | Methods Used for Paleobiological Reconstruction |
| --- | --- |
| †*Mitsukurina lineata* (Probst) | Isotopes ($\delta^{66}$Zn values) |
| †*Araloselachus cuspidatus* (Agassiz) | Isotopes ($\delta^{66}$Zn values), skeletal remains |
| †*Carcharoides catticus* (Philippi) | Artificial tooth set |
| †*Carcharias gustrowensis* (Winkler) | Skeletal remains |
| †*Carcharodon hastalis* (Agassiz) | Bite traces on fossil dolphin skeleton, tooth height and width, skeletal remains, stomach content, isotopes ($\delta^{66}$Zn and $\delta^{15}$N$_{EB}$ values) |
| †*Carcharodon hubbelli* Ehret, MacFadden, Jones, DeVries, Foster and Salas-Gismond | Vertebral centra, tooth height |
| †*Megalolamna paradoxodon* Shimada, Chandler, Lam, Tanaka & Ward | Tooth height and shape, paleoenvironment |
| †*Otodus* (*Megaselachus*) *megalodon* (Agassiz) and†*O.* (*M.*) *chubutensis* (Ameghino) | 2D and 3D reconstructions, isotopes ($\delta^{66}$Zn, $\delta^{18}$O$_P$, and $\delta^{15}$N$_{EB}$ values), vertebral centra, tooth height and width, plaeoenvironment, comparison with the extant great white shark (*Carcharodon carcharias*), placoid scales and tessellated calcified cartilage remains, marine mammal bones with bite traces from †*Otodus* teeth |
| †*Parotodus benedeni* (Le Hon) | Artificial tooth set, tooth shape and height, comparison with members of Lamnidae and Otodontidae |
| †*Keasius parvus* (Leriche) | Shape of gill rakers, skeletal remains |
| †*Megachasma applegatei* Shimada, Welton and Long | Tooth shape (including a landmark-based geometric morphometric analysis), paleoenvironment |
| †*Pachyscyllium distans* (Probst) and †*Pachyscyllium dachiardii* (Lawley) | Paleoenvironment |
| †*Hemipristis serra* Agassiz | Tooth size, artificial tooth set, paleoenvironment, isotopes ($\delta^{66}$Zn value) |
| †*Galeocerdo aduncus* (Agassiz) | Preserved jaw fragment, bite marks on a †*Metaxytherium* carcass and on a crocodilian coprolite, isotopes ($\delta^{66}$Zn value) |
| †*Physogaleus contortus* Gibbes | Tooth shape, teeth association with a cetacean carcass |
| †*Carcharhinus dicelmai* Collareta, Kindlimann, Baglioni, Landini, Sarti, Altamirano, Urbina & Bianucci | Tooth size, paleoenvironment |

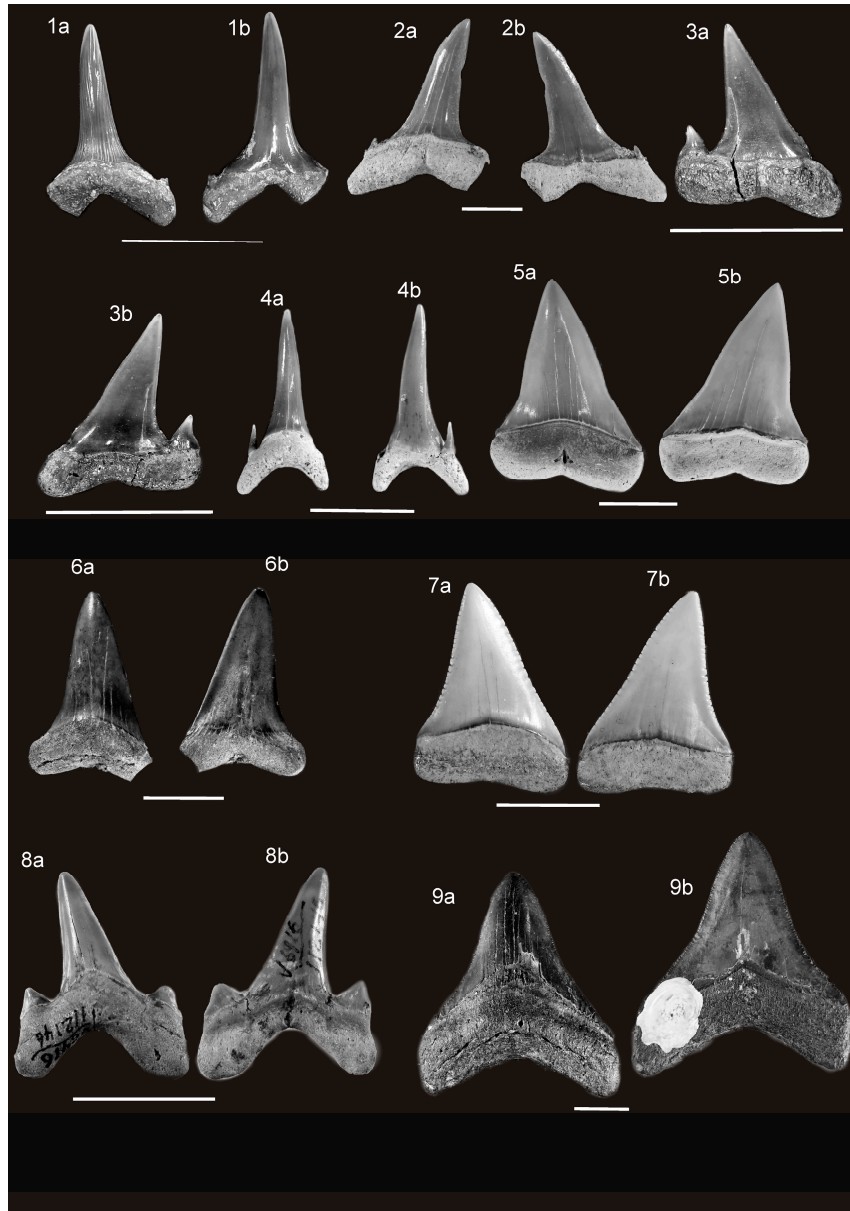

**Figure 2.** (**1a**,**1b**) *Mitsukurina lineata* (Probst). SMNS 97016/10, Miocene, Rengetsweiler, Baden-Württemberg, Germany. (**1a**) Lingual view; (**1b**) labial view. Scale: 10 mm. (**2a**,**2b**) *Araloselachus cuspidatus* (Agassiz). SMNS 97269, Miocene, Kühnring, Lower Austria. (**2a**) Lingual view; (**2b**) labial view. Scale: 10 mm. (**3a**,**3b**) *Carcharoides catticus* (Philippi). SMNS 97015/42, Miocene, Rengetsweiler, Baden-Württemberg, Germany. (**3a**) Lingual view; (**3b**) labial view. Scale: 10 mm. (**4a**,**4b**) *Carcharias gustrowenis* (Winkler). SMNS 97015/55, Miocene, Rengetsweiler, Baden-Württemberg, Germany. (**4a**) Lingual view; (**4b**) labial view. Scale: 10 mm. (**5a**,**5b**) *Carcharodon hastalis* (Agassiz). Broad-toothed morphotype. SMNS 97270, Miocene, Atacama Desert, Chile. (**5a**) Lingual view; (**5b**) labial view. Scale: 20 mm. (**6a**,**6b**) *Carcharodon hastalis* (Agassiz). "Narrow-toothed" morphotype. SMNS 55505, Miocene, Baltringen, Baden-Württemberg, Germany. (**6a**) Lingual view; (**6b**) labial view. Scale: 20 mm. (**7a**,**7b**) *Carcharodon hubbelli* Ehret, MacFadden, Jones, DeVries, Foster and Salas-Gismond. SMNS 97271, Miocene, Peru. (**7a**) Lingual view; (**7b**) labial view. Scale: 20 mm. (**8a**,**8b**) *Megalolamna paradoxodon* Shimada, Chandler, Lam, Tanaka & Ward. UCMP 112146, Miocene, Jewett Sand, Kern County, California, USA. (**8a**) Lingual view; (**8b**) labial view. Scale: 20 mm. Images courtesy of K. Shimada, used with permission. (**9a**,**9b**) *Otodus* (*Megaselachus*) *megalodon* (Agassiz). SMNS 97266, Miocene, Malta. (**9a**) Lingual view; (**9b**) labial view. Scale: 20 mm.

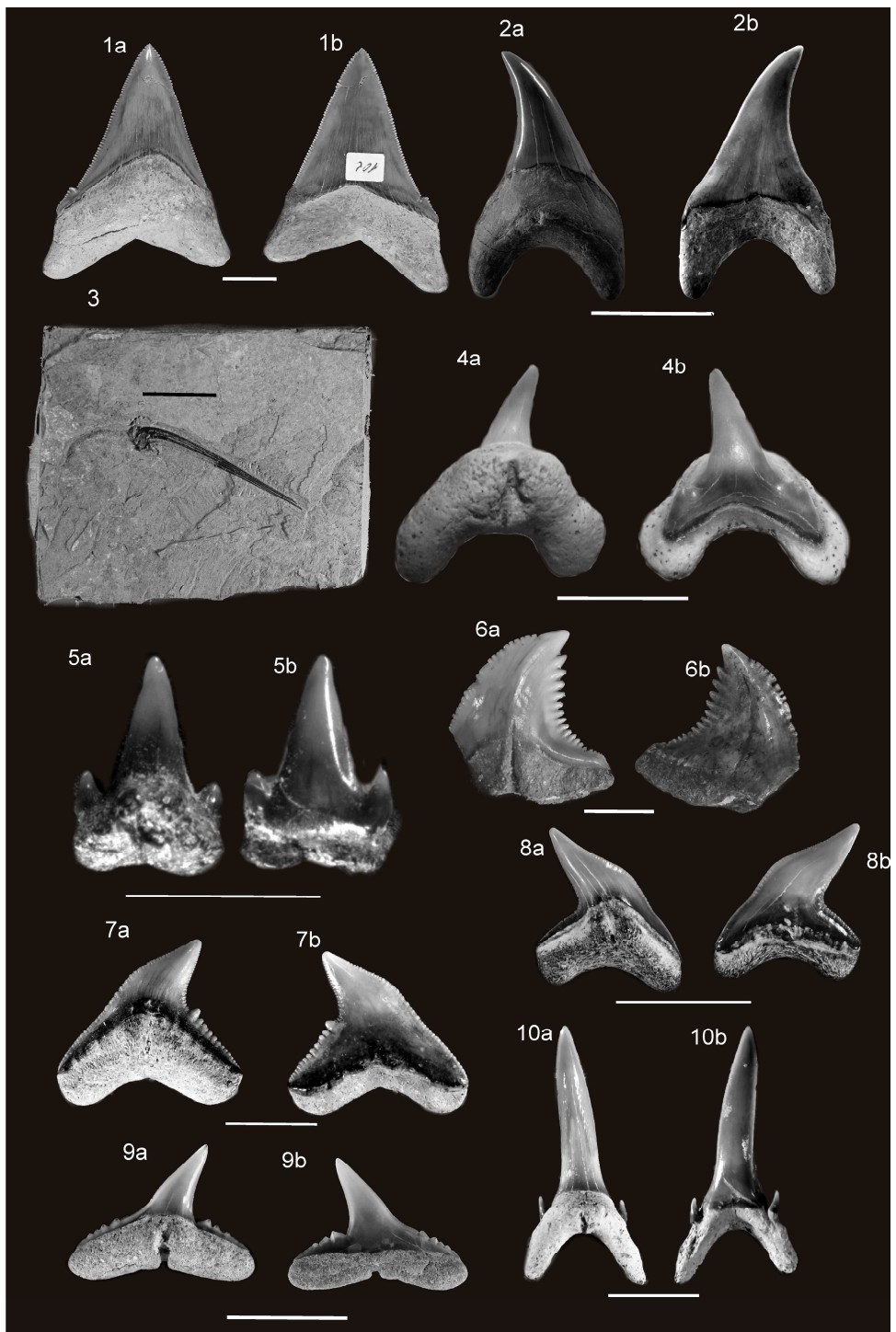

**Figure 3.** (**1a,1b**) *Otodus* (*Megaselachus*) *chubutensis* (Ameghino). SMNS 97267, Miocene, Lake Constance, Germany. (**1a**) Lingual view; (**1b**) labial view. Scale: 20 mm. (**2a,2b**) *Parotodus benedenii* (Le Hon). Miocene, Rengetsweiler, Baden-Württemberg, Germany. Specimen housed in a private collection. (**2a**) Lingual view; (**2b**) labial view. Scale: 20 mm. (**3**) *Keasius parvus* (Leriche). SMNS 80740/16, gill raker from the Bodenheim Formation, Oligocene. Rauenberg, Baden-Württemberg, Germany. Scale: 20 mm. (**4a,4b**) *Megachasma applegatei* Shimada, Welton and Long). LACM 122190, Miocene, Pyramid Hill Sand Quarry in southeastern San Joaquin Valley, California. Photos courtesy of Kenshu Shimada, used with permission. (**4a**) Lingual view; (**4b**) labial view. Scale: 5 mm. (**5a,5b**) *Pachyscyllium dachiardii* (Lawley). SMNS 56753, Miocene, Ursendorf, Baden-Württemberg, Germany. (**5a**) Lingual view; (**5b**) labial view. Scale: 5 mm. (**6a,6b**) *Hemipristis serra* Agassiz. SMNS

85944/1, Miocene, Baltringen, Baden-Württemberg, Germany. (**6a**) Lingual view; (**6b**) labial view. Scale: 10 mm. (**7a,7b**) *Galeocerdo aduncus* (Agassiz). SMNS 97268, Miocene, Rammingen, Baden-Württemberg, Germany. (**7a**) Lingual view; (**7b**) labial view. Scale: 10 mm. *Physogaleus contortus* (Gibbes). SMNS 97272, Miocene, Will Beach, Maryland, USA. (**8a**) Lingual view; (**8b**) labial view. Scale: 15 mm. (**9a,9b**) *Carcharhinus dicelmai* Collareta, Kindlimann, Baglioni, Landini, Sarti, Altamirano, Urbina & Bianucci. MUSM 4697, Miocene, Peru. (**9a**) Lingual view; (**9b**) labial view. Scale: 5 mm. Photos courtesy of Alberto Collareta, used with permission. (**10a,10b**) *Carcharias gustrowenis* (Winkler,). SMNS 97015/55, Miocene, Rengetsweiler, Baden-Württemberg, Germany. (**10a**) Lingual view; (**10b**) labial view. Scale: 10 mm.

## 4. Results

### 4.1. Paleobiology of Extinct Neogene Shark Species

Lamniformes Berg
Mitsukurinidae Jordan
✝*Mitsukurina lineata* (Probst) (Figure 2(1a,1b))

This is possibly the ancestor to the Recent *M. owstoni* (Jordan). Teeth of ✝*M. lineata* are found in bathyal and neritic deposits from the Early and Middle Miocene of Europe and South Korea [3,23]. $\Delta^{66}$Zn values for teeth from the Early Miocene of Baden-Württemberg, Germany, show a lifestyle similar to that of *Pseudocarcharias kamoharai* (Matsubara) (syn. *P. rigida*) [18]. The latter species feeds on bony fishes, squid, and shrimp [24], which is also the case for the Recent *M. owstoni* [24]. Although *M. owstoni* is a mostly bathyal shark, rarely occurring in shallow waters close to shore [24], the teeth of ✝*M. lineata* have also been found in neritic deposits, as mentioned above. The species possibly visited shallower waters in search of food or followed schools of fishes [25].

Odontaspididae Müller & Henle
✝*Araloselachus cuspidatus* (Agassiz) (Figure 2(2a,2b))

There are differing opinions as to whether this species belongs to the genus *Carcharias* (see the extant *Carcharias taurus*) or to the extinct genus ✝*Araloselachus* [3,11,26]. Likewise, its relationship with the species ✝*Araloselachus vorax* (Le Hon), which had similarly shaped teeth, is not yet resolved [3,16,26–28]. ✝*Araloselachus cuspidatus* is known from Miocene neritic deposits in Europe, North America, and Central Asia [3], as well as from older deposits of Oligocene age [11,29]. Its teeth are very abundant. They have a grasping, odontaspid shape but with a broader crown and often larger size than in ✝*Carcharias contortidens* or *C. taurus*. $\Delta^{66}$Zn values indicate that ✝*A. cuspidatus* was likely a higher-trophic-level piscivore than ✝*M. lineata* and *Pseudocarcharias kamoharai* (syn. *P. rigida*) [18], also supported by the larger tooth size of ✝*A. cuspidatus* [18]. A partial skeleton of ✝*A. cuspidatus*, including fetuses, from the Oligocene of Germany was illustrated and described [11]. An estimated body length of ca. 5 m was mentionedfor this specimen [11]. Adelophagy (intrauterine cannibalism), which is characterized by larger pups preying on smaller ones, is well documented among unborn pups of the extant *Carcharias taurus* [11,24,30]. This might also have occurred in ✝*A. cuspidatus* and could explain the large number of incomplete embryos recovered [11].

✝*Carcharoides catticus* (Philippi) (Figure 2(3a,3b)).

Two species of ✝*Carcharoides* are known from the Neogene. Using both of these species, an artificial tooth set was constructed for ✝*Carcharoides catticus* (Philippi) [31]. Based on the tooth morphology of ✝*C. catticus*, this species was considered to be a synonym of *Triaenodon obesus* (Rüppell) [32]; however, other authors dealing with this species do not share this opinion and cite this species as ✝*C. catticus* [28,31]. Dried jaws or teeth of *T. obesus* were not available to the authors for comparison; therefore, the fossil teeth are treated here as ✝*C. catticus*. The reconstruction of the dentition shows similarities to the dentition of *Carcharias* and *Odontaspis* [31]; therefore, a piscivorous diet can be also

assumed for **†***C. catticus*. Weak ontogenetic heterodonty for members of **†***Carcharoides* was mentioned [31]. The species **†***C. catticus* first appears in the Oligocene. and teeth of up to Middle Miocene age can be found in the neritic sediments of Europe and North America [31,32].

Carchariidae Müller & Henle, 1838
**†***Carcharias gustrowensis* (Winkler) (Figure 2(4a,4b)).
This species existed from the Oligocene to at least the Lower Miocene [10,28]. Hovestadt & Hovestadt-Euler (2010) [10] A partial skeleton of a gravid shark with eight fetuses, also associated with a myliobatoid tail spine and a chimaeroid dorsal fin spine was described from the Oligocene of Baden-Württemberg, Germany [10]. The variation in the length of the fin radials in **†***C. gustrowensis* resembles the pectoral fin skeleton of *Carcharias taurus* [10]. The myliobatoid and chimaeroid spines are likely remains of prey that have pierced the skin or cartilage of the jaw area. In addition to the Oligocene and Miocene of Germany, the species is also known from the Miocene of the Netherlands, Belgium, and Hungary, as well as the USA (North Carolina and Chesapeake Bay) [28,33].

Lamnidae Müller & Henle, 1838
The *Carcharodon* complex
The most recent systematic arrangement of the tooth shape indicates that **†***Carcharodon hastalis* (Early Miocene–Pleistocene) is the oldest member of this genus, followed by **†***Carcharodon hubbelli* (Late Miocene) and lastly the extant species *Carcharodon carcharias* (Early Pliocene–Recent) [8,9].
**†***Carcharodon hastalis* (Agassiz) (Figure 2(5a,5b,6a,6b))
Teeth of this species are common worldwide in temperate to tropical neritic deposits of Early Miocene to Pleistocene age [3,34]. The generic referral of this species remains debated. There are also some uncertainties at the species level, with a narrow-toothed morphotype as well as a broad-toothed one. There is therefore a discussion of whether two other broad-toothed species (**†***C. plicatilis* and **†***C. xiphodon*) (Figure 2(5a,5b)) are distinct from the narrow **†***C. hastalis* tooth morphotype (Figure 2(6a,6b)) [9,16,25,32,35,36]. This morphological difference could represent sexual dimorphism or ontogenetic change [9]. Assuming that all the referred teeth belong to only one species, the maximum body size would have been 6–7.6 m, with anterior teeth up to 8.1 cm in height [32]. A partially complete, articulated skeleton of a **†***C. hastalis* juvenile, including stomach contents, was documented from the Late Miocene of Peru [37]. The total body length of the immature specimen was estimated to be 2.3–2.4 m. The Meckel's cartilages are very similar to those of various extant Lamniformes (including *Carcharodon carcharias* and *Isurus* spp.). The teeth are distinctly more slender than the adult teeth of **†***C. hastalis*, in agreement with the pronounced ontogenetic heterodonty recognized in this species) [37]. The stomach contents consisted of fishes, including the pilchard *Sardinops* cf. *sagax.* It is possible that individuals with the narrow-toothed morphology had a piscivorous lifestyle, whereas those with the broad-toothed morphotype had a diet primarily consisting of small marine mammals [37]. In the Pisco Formation, sixteen teeth of **†***C. hastalis* were found in close contact with a balaenopterid whale skeleton [38]. A tooth of **†***C. hastalis*, early in its development, which was completely penetrated by a myliobatiform caudal spine, could be verified from the Calvert Cliffs (USA) (8–18 Ma, Miocene) [16]. Bite traces were found on a well-preserved fossil dolphin skeleton from the Pliocene of Italy [39]. Most bite traces were caused by a shark with unserrated teeth and about 4 m in length and were attributed to **†***C. hastalis* based on their morphology and the distribution of traces on the skeleton. Additionally, bite traces attributed to **†***C. hastalis* on cetacean skeletons from the Zanclean (Early Pliocene) of South Africa were described [15]. In contrast to the bite trace record, $\delta^{15}N_{EB}$ values in Miocene-aged **†***C. hastalis* teeth were similar to those of Pliocene and extant *C. carcharias*, but lower, more piscivore-like values were found in the Pliocene [17]. Congruently, $\delta^{66}Zn$ signals indicated that **†***C. hastalis* teeth from the Early Miocene of Malta had a higher trophic

position than teeth from the Early Pliocene of North Carolina. However, conspecific teeth from the Miocene of Baden-Württemberg, Germany, also indicated a lower trophic position, potentially indicating that the regional availability of different prey types influenced the diet [18]. The same result was recovered for †*Hemipristis serra* between the two Miocene localities, lending support to this hypothesis. However, another possible interpretation is that the previously mentioned tooth morphotypes were correlated with trophic signal. Based on material in collections, it seems that only the narrower morphotype was present in the Early Miocene of Baden-Württemberg [40]. Reasons underlying the extinction of †*C. hastalis* are unknown.

†*Carcharodon hubbelli* Ehret, MacFadden, Jones, DeVries, Foster, and Salas-Gismond (Figure 2(7a,7b)).

A well-preserved jaw containing 222 teeth and associated with a series of 45 vertebral centra was recovered from the Late Miocene Pisco Formation of Peru. The teeth show similarities to those of *C. carcharias* and †*C. hastalis*, and †*C. hubbelli* was therefore interpreted as an intermediate species between †*C. hastalis* and *C. carcharias* [8,9]. †*C. hubbelli* is also known form the Late Miocene of California, USA, and Chile [41,42].

The examination of the vertebral centra yielded an age of at least 20 years. Based on measurements of the teeth and vertebral centra, this specimen is estimated to have had a minimum total body length of 4.80–5.07 m. The growth of †*C. hubbelli* appears to have been slower than that of Recent great white sharks [8,9]. †*C. hubbelli* fed on marine mammals [9].

†Otodontidae Glickman, 1964.

†*Megalolamna paradoxodon* Shimada, Chandler, Lam, Tanaka, & Ward (Figure 2(8a,8b)).

This species is known from teeth from the Early Miocene of the USA (North Carolina, California), Japan, and Peru [43,44], as well as from Baden-Württemberg, Germany (as "*Lamna* sp.") [25]. All the deposits represent shallow-water shelf-type coastal environments [25,43,44]. The largest came from an individual measuring at least 3.7 m in total length [43]. Based on the shape of the anterior and lateral teeth, the diet of †*M. paradoxodon* may have included relatively large prey, such as medium-sized (ca. 0.5–1 m) fishes, captured with the anterior teeth and cut by the distal portion of the dentition to a size suitable for ingestion [43].

†*Otodus* (*Megaselachus*) *megalodon* (Agassiz) (Figure 2(9a,9b)) and †*O.* (*M.*) *chubutensis* (Ameghino) (Figure 3(1a,1b)).

In the past, these extinct species have been placed in various genera (*Carcharodon*, †*Procarcharodon*, †*Carcharocles*, †*Megaselachus*); they are currently placed in †*Otodus*, and †*Megaselachus* is considered to be a subgenus [3,16].

†*Otodus* is divided into two chronospiecies: †*O.* (*M.*) *chubutensis*, with lateral cusplets or only traces thereof, and †*O.* (*M.*) *megalodon*, which lacks lateral cusplets. In Early Miocene deposits, teeth with cusplets are more abundant than uncuspleted ones. Moving upwards through the Miocene profile, uncuspleted *Otodus* teeth increase in relative abundance and the cuspleted ones eventually disappear entirely [45] (pers. observ. O.H.). A definitive separation between all the teeth of the taxa †*O. chubutensis* and †*O. megalodon* is impossible, because a complex mosaic evolutionary continuum characterizes this transformation, particularly in the loss of the lateral cusplets [45]. The cuspleted and uncuspleted teeth of †*Otodus* (*Megaselachus*) spp. are therefore designated as chronomorphs, because there is broad overlap between them both morphologically and chronologically. The †*O. chubutensis/megalodon* problem was discussed in detail in the literature [16,33,45,46].

The large, triangular teeth of *Otodus* spp. are surely the most easily recognizable shark teeth. †*Otodus* teeth are found worldwide in neritic deposits of the Neogene Epoch (see Cappetta 2012) [40]. The teeth of †*O.* (*M.*) *chubutensis* can reach a height of 13 cm; the ones from †*O.* (*M.*) *megalodon* can reach 17 cm [33]. Based on the tooth size, the maximum body length of †*O.* (*M.*) *megalodon* was probably between 18 and 20 m [47]. Individuals of †*O* (*M.*)

*megalodon* were, on average, larger in cooler waters than those living in warmer waters [47]. In the shallow marine Miocene Gatún Formation of Panama, the majority of ✝*O. (M.) megalodon* teeth are very small [48]. The individuals from Gatún were mostly juveniles and neonates, with estimated body lengths of between 2 and 10.5 m. They therefore proposed that the Gatún Formation represents a paleo-nursery area for ✝*O. (M.) megalodon* [47]. Based on statistical analyses, the presence of five potential nurseries were noted, ranging from the Langhian (Middle Miocene) to the Zanclean (Pliocene) in age, with higher densities of individuals with estimated body lengths within the range typical of neonates and young juveniles [49]. However, it was argued by other authors that, while it is possible that neonatal ✝*O. (M.) megalodon* could have utilized nursery areas, the previously identified paleo-nurseries may reflect temperature-dependent trends rather than inferred life history strategies [46].

A viviparous reproductive strategy characterized by matrotrophy via oophagy is primitive for crown-lamniform sharks [50], resulting in large body sizes at birth. This is consistent with the inferred life history of ✝*O. (M.) megalodon* [51]. Incremental growth bands in the fossil vertebrae of a 9.2-m-long individual from the Miocene of Belgium (see below) reveal that the shark was born large at 2 m in length and that this specimen died at age 46 [50]. It was estimated that ✝*O. (M.) megalodon* had a lifespan of at least 88–100 years and that it had a slightly higher growth rate (19–23 cm/year) during the first 7 years of life relative to the remainder of its life (11–18 cm/year) [51]. Tessellated calcified cartilage remains beside the teeth of a ca. 11.7-m-long individual could be verified from the Miocene of Japan [21]. The morphology of each tessera (i.e., predominantly hexagonal) and the arrangement of tesserae as a tessellated calcified cartilage sheet in ✝*Otodus* (*M.*) *megalodon* are virtually identical to those of extant chondrichthyans [21]. Further, it was found that the size range of tesserae observed in the estimated 11.7-m-long individual of ✝*O. (M.) megalodon* is comparable to that of extant chondrichthyans, indicating that a larger body size does not necessarily produce larger tesserae [21]. This observation suggests that, in ✝*O. (M.) megalodon*, as in extant sharks, skeletal elements sheathed by tesserae developed through biomineralization along the margins of existing tesserae to form new tesserae, despite its gigantic body size [21]. The first reconstruction of the skeletal anatomy of ✝*Otodus* was performed in the year 1996 [52]. The most recent anatomical reconstructions were developed in 2020 and 2022 [19,20]. In 2020 a two-dimensional reconstruction of ✝*O. megalodon,* was produced, based on comparisons with extant Lamniformes [19]. The results suggest that a 16 m ✝*O. (M.) megalodon* likely had a head ~4.65 m long, a dorsal fin ~1.62 m tall, and a tail ~3.85 m high [19]. In 2022, a three-dimensional model of ✝*O. megalodon* was published [20]. The basis for the model was a vertebral column with 141 centra, belonging to a single, 9.2-m-long individual housed in the Royal Belgian Institute of Natural Sciences in Brussels, Belgium, in addition to comparisons with the skeleton of the Recent great white shark, *Carcharodon carcharias* [20]. This vertebral column was recovered from the Antwerp Basin in the 1860s; however, neither the locality nor an age has been specified for the specimen beyond a Miocene range (23 to 5.3 Ma) [20]. The reconstruction yielded a total length of 15.9 m and a body mass of 61,560 kg. The gape size was determined at different angles: the gape was 1.2 m in height at a 35° angle and 1.8 m in height at 75° angle. The gape was 1.7 m in width at both angles. The stomach volume was estimated to have been 9605 L. Prey up to 8 m in length could have been ingested whole, whereas larger prey (e.g., prey the size of the extant humpback whale, *Megaptera novaeangliae*) would have required additional processing [20]. It was calculated that the modeled ✝*O. (M.) megalodon* had an energy requirement of 98,175 kcal per day. Additionally, the mean absolute speed for the model was calculated at 1.4–4.1 m/s (=ca. 5.0–14.8 km/h) and the mean relative cruising speed was estimated at 0.09 body lengths per second [20]. However, other authors estimated lower cruising speeds for ✝*O. (M.) megalodon*, at 2.0 km/h, with a range of 0.9–3.0 km/h, based on the morphology of its placoid scales [21]. This authors also found that the general size of the placoid scales in the vast majority of extant pelagic lamniforms and carcharhiniforms, as well as in extinct lamniform taxa such as ✝*Cretoxyrhina*, ✝*Cretodus*,

and †*Squalicorax*, was similar to the overall scale size of the much larger †*O. megalodon*. This demonstrates that the exceptionally large body sizes seen in †*O.* (*M.*) *megalodon* did not result in exceptionally large placoid scales [21]. Rather, new placoid scales of a similar small size were added throughout the ontogeny as the shark grew [21]. All the authors used the chronospecies name †*O. megalodon*, but there is no reason to assume that these data cannot be extrapolated to †*O. chubutensis* if of similar size.

†*Otodus* spp. were the top predators during the Miocene and Early Pliocene. There are many examples of marine mammal bones with bite traces from †*Otodus* teeth, including, e.g., small-sized mysticete cetaceans and pinnipeds from the Upper Miocene Pisco Formation (Southern Peru) [53] and a mysticete caudal vertebra from the Pliocene of North Carolina [54]. However, in the majority of cases, it remains unclear whether these feeding events on mammals document active hunting or scavenging [18]. Using enameloid-bound $\delta^{15}N$ ($\delta^{15}N_{EB}$) in †*Otodus* teeth, it could be determined that †*Otodus* (*M.*) *megalodon* as well as †*O.* (*M.*) *chubutensis* occupied a higher trophic level than that of any known marine species, extinct or extant [17]. The $\delta^{15}N_{EB}$ values show a large range for †*O.* (*M*). *megalodon*, which may reflect a generalist diet, with individuals feeding across many prey types and different trophic levels [17]. Many extant apex predatory sharks are also opportunistic in their prey selection [18]. Despite the bite traces on the mysticete bones noted above, the high $\delta^{15}N_{EB}$ values indicate that baleen whales were not the dominant prey of †*O. megalodon*, as extant baleen whales have a low trophic level and a correspondingly low $\delta^{15}N$ (Kast et al., 2022) [17]. $\delta^{66}Zn$ values derived from the tooth enameloid of †*O. megalodon* were used to find support for the previous conclusion that †*Otodus* spp. were apex predators feeding at a very high trophic level [18]. However, during the Early Pliocene, the †*Otodus* lineage represented by †*O.* (*M*). *megalodon* showed a considerable increase in mean $\delta^{66}Zn$ values in Atlantic populations, hinting at a reduced trophic position for the megatooth shark lineage in the Atlantic [18]. This could indicate a dietary shift, specifically that lower-trophic-level mammalian prey such as mysticetes (and perhaps herbivorous sirenians) may have become an important dietary component for Atlantic populations of †*O.* (*M*). *megalodon*. Extinct small- and medium-sized mysticetes (e.g., Cetotheriidae and various small balaenids and balaenopterids) were abundant during the Early Pliocene and were thus available as prey for *Otodus* spp. [18]). As can be seen, the isotopic results are partially in conflict with respect to the trophic level.

Thermophysiology is another important area of investigation concerning the paleobiology of the Neogene †*Otodus* spp. The question of endothermy in Neogene †*Otodus* sharks were examined using $\delta^{18}O_p$ values (P = phosphate) [55]. The measurements support endothermy in †*Otodus* (*M.*) *megalodon* and †*O.* (*M.*) *chubutensis* [55]. Based on their lower estimates of the cruising speed, it was suggested that the function of regional endothermy shifted from maintaining high cruising speeds to accelerating digestion and nutrient absorbtion during the evolution of gigantism in otodontids [21].

Regarding the extinction of †*Otodus* (*M.*) *megalodon*, two dates are reported in the newer literature: (1) before c. 2.6 Ma (Pliocene/Pleistocene boundary) [56]; (2) before c. 3.6 Ma (Early–Late Pliocene boundary) [57]. There are different opinions regarding the importance of competition with great white sharks or the extinction of small to mid-sized mysticete prey species, as possible drivers of the extinction [17,18,58]. Competition with hypercarnivorous odontocetes may have also played a role in the extinction process [18,57]. Concerning climatic changes as potential causes of the extinction, no evidence was found for direct effects of the global temperature [58]. It was noted that the gigantic body size, when combined with with the high metabolic cost of maintaining an elevated body temperature, may have made †*Otodus* species more vulnerable to extinction than the sympatric sharks that survived the Pliocene Epoch [55]. In summary, the reasons for the extinction of †*O.* (*M.*) *megalodon* are still unknown.

†*Parotodus benedenii* (Le Hon) (Figure 3(2a,2b)).

The teeth of †*Parotodus benedenii* can be up to 6 cm high. This species has been widely reported from the Early Oligocene through the Early Pliocene fossil beds of Europe (Belgium, Germany, Hungary, Italy, Malta, the Netherlands, Portugal, Slovakia, and Switzerland), Africa (Angola and South Africa), the Azores, and the United States, as well as from Australia, Japan, and New Zealand in the Western Pacific [16]. Despite its broad geographical distribution, this species is rare in Neogene deposits. During the Neogene, a clear increase in tooth size occurred, accompanied by a notable thickening of the root, which became very stout and globular [3]. Different authors illustrated an artificial tooth set of this species [32,33,59]. †*P. benedenii* was reconstructed as a large, hypercarnivorous shark that inhabited pelagic settings and fed primarily on large, soft prey and scavenged items [60]. Thus, some ecological partitioning did likely exist between †*P. benedenii* and other elasmobranch apex predators (including the extant species *Carcharodon carcharias*, *Carcharhinus leucas*, and *Galeocerdo cuvier* during the Pliocene) in Neogene mid-latitude seas. The body length of †*P. benedenii* was estimated at over 7 m [60] or between 6 and 7.5 m [32].

Cetorhinidae Gill
†*Keasius parvus* (Leriche) (Figure 3(3)).
This species was originally placed in the basking shark genus *Cetorhinus*. In 2013, the species was placed in the newly erected genus †*Keasius* [61], based on the shape of the gill rakers, the vertebral centra, and the dentition. †*K. parvus* existed from the Middle Eocene to Middle Miocene [62]. Remains have been found in Europe, Mexico, and Japan [61]. A partial skeleton of †*K. parvus* was described from the Oligocene (Rupelian) of Germany [62]. †*K. parvus* possessed a filter feeding apparatus similar to that of the extant *Cetorhinus maximus*, and it can be assumed that the species shared the same feeding habits. The aforementioned skeleton came from a ca. 2-m-long animal [62]. The maximum length of †*K. parvus* is estimated at 4.5–5 m [62].

Megachasmidae Taylor, Compagno & Struhsaker
†*Megachasma applegatei* Shimada, Welton and Long, 2014 (Figure 3(4a,4b)).
The teeth of this extinct megamouth shark are known from Late Oligocene–Early Miocene marine deposits of the Western USA [63]. †*M. applegatei* could have measured approximately 6 m in total length and likely had a broad diet, possibly including small fishes and planktonic invertebrates. The fossil record indicates that †*M. applegatei* either had a broad bathymetric tolerance or was a nektopelagic feeder over both deep- and shallow-water habitats [64]. An artificial tooth set of this species was examined via landmark-based geometric morphometric analysis [63]. The teeth were more variable in shape than those of the extant *Megachasma pelagios* (Taylor, Compagno, & Struhsaker). The teeth of the fossil species were probably arranged in the typical heterodont "lamnoid tooth pattern" [65] as in predatory lamniform sharks.

Carcharhiniformes Compagno
Scyliorhinidae Gill
†*Pachyscyllium distans* (Probst) and †*Pachyscyllium dachiardii* (Lawley) (Figure 3(5a,5b)).
Both catshark species lived contemporaneously and their teeth are widespread in the Miocene and Early Pliocene of Europe (e.g., Germany, Belgium, France, Netherlands, Portugal, Italy) [28,40,66]. Both species had very similar teeth; therefore, only a tooth from *P. dachiardii* was illustrated. The only known information regarding the paleoecology of these taxa is that both were thermophilic sharks) [28,66].

Hemigaleidae Hasse
†*Hemipristis serra* Agassiz, 1843 (Figure 3(6a,6b)).
The species is very widely distributed from the Late Oligocene (Chattian) through the Pleistocene in formations representing warmer-water regions of the Atlantic Ocean, Caribbean Sea, Mediterranean Sea, Indian Ocean, and Pacific Ocean [16]. An artificial

tooth set for this species was published [32]. Whether **†***H. serra* is the direct ancestor to the Recent *H. elongata* (Klunzinger) is questionable. Based on histological differences between its teeth and those of the extant *H. elongata* (Klunzinger, 1871), it was suggested that the generic reassignment of **†***H. serra* is warranted [67]. **†***H. serra* probably reached a length of c. 6 m [68], whereas the Recent species only attains lengths of 2.3–2.4 m [30]. There are some differences in tooth size through time and space. Teeth from the Early Miocene of Southern Germany have a maximum size of 31 mm in height and 25 mm in width [69], but teeth from the Early Pliocene of North Carolina, USA, reached a height of 41 mm and a width of 43 mm [32].

Based on the $\delta^{66}$Zn composition, **†***H. serra* from the Early Miocene of Malta occupied a higher trophic position than individuals from the Early Miocene of Baden-Württemberg, Germany. This is the same relative result recovered for individuals of **†***Carcharodon hastalis* between the two localities; different prey availability or a shorter trophic chain in the German Molasse Basin may also be driving the observed pattern in this case. The Maltese specimens have a similar trophic position to **†***Galeocerdo aduncus* [18].

Galeocerdonidae Poey
**†***Galeocerdo aduncus* (Agassiz) (Figure 3(7a,7b)).
This ancient tiger shark is found worldwide in neritic sediments of Oligocene to Late Miocene age [70]. A preserved jaw fragment from the Miocene (8 to 18 Ma) of Calvert Cliffs, USA was illustrated [16]. The teeth are similar to those of the extant tiger shark *G. cuvier*, apart from differences concerning the serration as well as the size [70]. **†***G. aduncus* teeth are smaller. However, some authors [32] placed this species in synonymy with the extant *G. cuvier* on the basis of similarities in overall morphology.

Fossil evidence from the Middle Miocene of the Styrian Basin (Austria) shows that **†***G. aduncus* fed on a sirenian carcass (**†***Metaxytherium* sp.) [71]. Other authors were also able to match tooth marks on a crocodilian coprolite to this species [72]. Zinc isotope values in the *Galeocerdo* lineage show no statistical variability with either age or locality, suggesting that tiger sharks occupied a similar trophic level and ecological role in the marine ecosystem since at least the Early Miocene [18]. **†***G. aduncus* likely had a similar lifestyle to that of the extant *G. cuvier*, despite having smaller teeth.

**†***Physogaleus contortus* (Gibbes) (Figure 3(8a,8b)).
Teeth are known from the Early and Middle Miocene of the Eastern United States (Maryland, North Carolina, and Virginia), Cuba, Panama, Peru, Germany, and Hungary [16]. The paleobiology of **†***P. contortus* is largely unknown, although the slender, twisted tooth crowns are consistent with a largely piscivorous diet [16]. A sperm whale skeleton from the lower Calvert Formation of Popes Creek, Maryland, USA (Early to Middle Miocene) was associated with 37 **†***P. contortus* teeth [16]. Although the teeth are exceptionally large, these sharks were far too small to have attacked and killed such substantial prey. Typically, such an association of teeth would be attributed to scavenging, although this is difficult to confirm. Based on the tooth morphology, it seems equally plausible that this tooth concentration represents **†***Physogaleus* preying on small scavenging fishes attracted by the carcass [16].

Carcharinidae Jordan & Evermann
**†***Carcharhinus dicelmai* Collareta, Kindlimann, Baglioni, Landini, Sarti, Altamirano, Urbina, & Bianucci (Figure 3(9a,9b)).
This newly described species is known from the Lower Miocene Chilcatay Formation of Peru (type locality) and from the Lower- to mid-Miocene (Burdigalian to Lower Langhian) Cantaure Formation of Venezuela. The latter locality suggests a trans-Panamanian distribution for this ancient species [73]. Given the dimensions of its teeth, **†***C. dicelmai* was likely a diminutive carcharhinid and may have relied on small prey items (including, e.g., small bony fishes and invertebrates) that were individually captured and ingested through feed-

ing actions that involved clutching [73]. **†***C. dicelmai* may also have been an essentially thermophilic and very littoral shark [73].

Additional comments regarding fossil *Carcharhinus*: In the Pliocene of Tuscany, Italy, a fossil cetacean rib pierced by a partial requiem shark tooth (*Carcharhinus* sp.) was found [74]. Evidence for *Carcharhinus* sharks (mostly broad-toothed members of the genus) foraging upon cetaceans is preserved in the Mediterranean Pliocene fossil record in the form of bite traces and teeth associated with bones [74]. Species-level identifications were not provided.

*4.2. The Fossil Records of Extant Shark Species*

Hexanchiformes de Buen
Hexanchidae Gray
*Hexanchus griseus* (Bonnaterre)
Fossils of very large *Hexanchus* teeth (at least 25 mm in width) have been widely, if rarely, collected from Early Miocene to Pliocene sediments in Belgium, Chile, Italy, Japan, Malta, Peru, Portugal, and Spain, as well as California and North Carolina in the USA [16]. These were named as **†***Hexanchus gigas* (Sismonda) by Kent [16] or as *Hexanchus* sp. by Purdy et al. [32]. Apart from the large size, they are similar to the teeth of the extant *H. griseus*. As yet, it is unclear whether they represent separate species or are conspecific.

A large *Hexanchus* tooth was associated with a cetacean skeleton (**†***Cephalotropis coronatus* Cope) from the Late Miocene of Maryland, although it is uncertain whether this represents active predation or scavenging. Shark bite traces on a sirenian skeleton from Pliocene shoreface deposits of Tuscany (Italy) were mentioned, which can probably be attributed to an immature *H. griseus* [75].

*Notorynchus cepedianus* (Péron)
The fossil record of this extant species is not clear. Teeth of similar shape to those of *N. cepedianus* can be found from the Late Oligocene (Chattian) through the Late Miocene of Florida, Maryland, North Carolina, and Virginia, as well as Australia, Austria, the Azores, Belgium, Denmark, France, Germany, Japan, the Netherlands, Poland, Portugal, Slovakia, Spain, and Switzerland [16]. These fossil teeth have mostly been named as **†***Notorynchus primigenius* (Agassiz) [40]. There are, however, differing opinions regarding whether **†***N. primigenius* represents a distinct species [16] or is a synonym of *N. cepedianus* [32]. Interestingly, the geographic distribution of Recent *N. cepedianus* is quite unlike that of *Notorynchus* in the Neogene, with Recent members of this genus generally restricted to cool temperate waters, whereas, in the Neogene, the genus was also widely distributed in warm temperate and tropical waters [28].

*Heptranchias perlo* (Bonnaterre)
Fossil record: Early Miocene: Costa Rica [76]; Middle Miocene: Italy (Abruzzo, Parma) [77,78]; Late Miocene: Panama (Northern Panama) [79]; Portugal (Lisbon) (as "cf.") [80]; Late Miocene to Early Pliocene: Venezuela (Northeastern Venezuela) [81].

Squaliformes Goodrich
Centrophoridae Bleeker
*Centrophorus granulosus* (Bloch & Schneider)
Fossil record: Early to Middle Miocene: France (Vaucluse) [82]; Pliocene: Italy (Tuscany, Piedmont) and France (Le-Puget-sur-Argens) [83–85]. In the Miocene deposits of Europe and South America, many teeth have been named as *Centrophorus* cf. *granulosus* [86,87] since they show similarities to the extant *C. granulosus*. However, the dentition of the other 10 extant *Centrophorus* species is insufficiently known [22]. The assignment of isolated *Centrophorus* teeth to species is therefore not without problems.

*Deania calcea* (Lowe)

Fossil record: Early to Middle Miocene: France (Vaucluse) [82], Middle Miocene: Spain (Southeastern Spain) [88], Japan (Nagano Prefecture) (as "cf.") [89]; Early Pliocene: Italy (Parma) (as "cf.") [90].

Dalatiidae Gray
*Dalatias licha* (Bonnaterre)
Fossil record: Miocene: Italy (Sardinia) [91]; Early to Middle Miocene: France (Vaucluse, Southern France) [82,92,93], Colombia (Guajira Peninsula) (as "cf.") [94,95]; Middle Miocene: South Korea [23]; Early Miocene to Early Pliocene: Japan [96–99]; Late Miocene: Panama [79]; Pliocene: Japan [100]; Early Pliocene: France (Le-Puget-sur-Argens) [84]; Late Pliocene: Italy (Tuscany) [85].

Somniosidae Jordan
*Scymnodon ringens* du Bocage & Capello
Fossil record: Early Pliocene: Italy (Parma) [101]; Middle Pliocene: Italy (Romagna Apennines) (as "cf.") [102].

*Somniosus rostratus* (Risso)
Fossil record: Early Pliocene: Italy (Parma) [103].

*Zameus squamulosus* (Günther)
Fossil record: Early Pliocene: Italy (Parma) [101].

Orectolobiformes Applegate
Rhincodontidae Garman
*Rhincodon typus* Smith
Fossil record: Late Oligocene: USA (South Carolina) (as "cf.") [104]; Early Miocene: France (Occitania) (as *Rhincodon* sp.) [105]; Early to Middle Miocene: USA (Maryland, North Carolina) [32,106]; Late Miocene–Early Pliocene: Costa Rica [107].

Lamniformes Berg
Cetorhinidae Gill
*Cetorhinus maximus* (Gunnerus)
Fossil record: Following Hovestadt & Hovestadt-Euler [62], this extant species first appeared in the Middle Miocene, whereas Welton [108] cited the Late Miocene as the earliest occurrence. Material has been referred to this taxon from the Early to Middle Miocene: Japan (Saitama) [109]; Middle Miocene: Czech Republic (Kienberg) [110]; Late Miocene: USA (Oregon) (as "cf.") [108], USA (California) [111]; Late Miocene: Germany (Sylt, Lower Saxony [112,113]; Late Miocene to Early Pliocene: Chile (El Rincón) [114], Netherlands (Winterswijk-Almelo) [115]; Early Pliocene: Belgium (Kallo) [116], France (Le-Puget-sur-Argens, Anvers) [84,117]; Late Pliocene: Italy (Tuscany) [85].

Carchariidae Müller & Henle
*Carcharias taurus* Rafinesque
Teeth similar in shape to those of the extant *Carcharias taurus* (Rafinesque) can be found worldwide in Neogene neritic deposits. Teeth of this morphology are the most abundant shark teeth in these deposits and often occur en masse. Historically, Miocene teeth of this type have been identified as †*C. contortidens* (Figure 3(10a,10b)), but the relationship of this taxon with *C. taurus* is not completely clear [28]. Similar teeth from the Early Pliocene have been named as *C. taurus* [18,32]. One problem is that, despite their abundance, the teeth are often not completely preserved and therefore important details (e.g., lateral cusplets) are often missing.

Based on the $\delta^{66}$Zn values, *Carcharias* teeth show a relatively stable trophic level and ecological niche through time and space [18], and a similar lifestyle to that of the extant

*C. taurus* can be assumed for the Miocene representatives, despite the controversial species-level classification. Details of the biology of *C. taurus* can be found in Ebert et al. [30]. Today, this species is distributed in nearly all warm and tropical waters apart from the Eastern and Central Pacific [30]. During the Miocene and part of the Pliocene, members of the genus *Carcharias* (probably *C. taurus*) also occupied waters off the western coast of South America, where, today, the species is absent [118]. The latter authors suggested that the local extinction of *Carcharias* was the consequence of a drop in global temperatures during the Middle Pliocene and Pleistocene, accompanied by a coeval drop in sea level that reduced the shelf area and therefore the suitable habitat for this species. The establishment of the Panamanian isthmus prevented the later migration of *C. taurus* from the north [118].

Odontaspididae Müller & Henle
*Odontaspis ferox* (Risso)
Fossil record: Early Miocene: Chile (Central Chile); Middle Miocene: USA (North Carolina) [32,119]; Middle Miocene–Pliocene: Chile (Northern Chile) [42]; Late Miocene–Early Pliocene: Venezuela [81]; Early Pliocene: USA (North Carolina) [32]; Late Pliocene: Italy (Tuscany) [85].

Pseudocarchariidae Taylor, Compagno & Struhsaker
*Pseudocarcharias kamoharai* (Matsubara)
Fossil record: Early Miocene: Germany (Baden-Württemberg, Bavaria) [40,120], Austria (Upper Austria) [120], Hungary [121], Switzerland (Schaffhausen) [122]; Middle Miocene: Italy (Parma) [123]; Late Miocene: Portugal (Alvalade) (as "cf.") [124]; Late Miocene–Early Pliocene: Venezuela [81].

Alopiidae Bonaparte
*Alopias superciliosus* Lowe
Fossil record: Oligocene: Germany (Bavaria) (as "cf.") [125]; Early Miocene: USA (North Carolina) [126], Peru [44], Colombia as "cf.") [94]; Early Miocene to Early Middle Miocene: Japan [96]; Middle Miocene: Netherlands [127]; Middle Miocene to Early Pliocene: USA (Florida) [128]; Late Miocene: Panama [79,129], Portugal (Alvalade Basin, Lisbon) (as "cf.") [130,131]; France (Luberon) (as "cf.") [93]; Late Miocene–Early Pliocene: Venezuela, Costa Rica [81,107]; Pliocene: Italy (Tuscany) [132].

*Alopias vulpinus* (Bonnaterre)
Fossil record: Miocene: Myanmar [133], India (Orissa) [134]; Early Miocene: Portugal (Algarve) [135]. There are also many occurrences of this taxon in the literature with "cf." or "aff." originating from deposits dating from the Oligocene [44,104,130,136]. The fossil record of *A. vulpinus* therefore requires reassessment.

Lamnidae Müller & Henle
*Lamna nasus* (Bonnaterre)
Fossil record: Late Miocene: Netherlands (Liessel) [137]; Early Pliocene: Belgium (Kallo) [116]; Late Pliocene Italy (Tuscany) [138].

*Isurus oxyrinchus* Rafinesque
This species is noted in sediments dating from the Oligocene [136]. It is known from many deposits in Germany, Belgium, France, Italy, Switzerland, USA, Japan, Chile, and Africa [3]. Fossil teeth similar in shape to the extant *I. oxyrinchus* have sometimes been named as †*Isurus desori* (Agassiz) [69]. At the moment, it is not clear if †*I. desori* is a valid species or is a synonym of *Isurus oxyrinchus*.

*Isurus paucus* Guitart-Manday

Fossil record: Early Miocene to Early Middle Miocene: Japan (Central Japan) [139]; Middle Miocene–Pliocene: possibly Chile (Northern Chile) [42].

*Carcharodon carcharias* (Linnaeus)
The extant great white shark first appeared in the Miocene or Early Pliocene [3,16]. For details on the biology of the extant *C. carcharias*, see Domeier [140]. The teeth occur worldwide in neritic sediments. In a few cases, predatory or scavenging behavior of fossil *C. carcharias* has been documented in the fossil record, and, as with observations on the extant *C. carcharias*, cetaceans were important prey species [15,16]. Cigala-Fulgosi [141] described the skeleton of an extinct dolphin with bite traces attributed to *C. carcharias* from the Pliocene of Italy (Piacenza). To date, there are no studies documenting piscivory in *C. carcharias* in the fossil record [16]. The $\delta^{66}$Zn isotopic results indicate an increase in the trophic position of *C. carcharias* from the Early Pliocene to the Recent [18]. In a comparison between Recent and fossil *Carcharodon carcharias*, both mysticete and odontocete cetaceans appear to have been equally represented in the diet of this species during the Pliocene. In contrast, extant great white sharks primarily attack small odontocetes and only rarely attack mysticetes. This change could be due to both the general reduction in the body size of great white sharks over time, as well as the diminished diversity of the cetacean assemblage [142]. A sample of fossil teeth from Spain indicates that large *C. carcharias* close to 7 m long or larger were relatively common in the Early Pliocene [143]. Villafaña et al. [144] described a paleo-nursery area of the great white shark from the Pliocene of Chile. Fossil teeth of *C. carcharias* can often be found in the same deposits as the extinct megatooth shark *Otodus* (*Megalselachus*) *megalodon*—for example, in the Late Miocene/Early Pliocene of Chile [114]. This suggests that both sharks co-existed [143]. However, no direct interaction or competition between these two apex predators has been documented.

Carcharhiniformes Compagno
Triakidae Gray
*Galeorhinus galeus* (Linnaeus)
Fossil record: Late Eocene: USA (North Carolina) [145]; Early Miocene: USA (North Carolina) [126]; Late Miocene: Panama (as "cf.") [79]; Late Miocene–Early Pliocene: Chile (Bahía Inglesa) [114]; Early Pliocene: South Australia (as *Galeorhinus* cf. *australis*) [146]; Late? Pliocene: USA (California) (as *Galeorhinus zyopterus*) [147]; Late Pliocene: Chile (Valparaíso) [148].

Galeocerdonidae Poey
*Galeocerdo cuvier* (Péron & Lesueur)
Fossil record: Early Miocene: India (Gujarat) [149]; Middle Miocene: Hungary (Nyirád) [150], USA (Florida) [70]; Middle Miocene–Middle Pliocene: [151]; Late Miocene: Panama (Lago Bayano) [129]; Late Middle to Early Late Miocene: Panama (Central Panama) [152]; Late Miocene: Borneo (Brunei Darussalam) [153]; Pliocene: USA (Florida, North Carolina) (Webb & Tessmann 1968; Maisch et al., 2018) [154,155], Angola [156]; Early Pliocene: Libya [157]; Late Early/Early Late Pliocene: Italy (Tuscany) [158].

Carcharinidae Jordan & Evermann
*Carcharhinus amblyrhynchoides* (Whitley)
Fossil record: Late Miocene: Borneo (Brunei Darussalam) [159].

*Carcharhinus amblyrhynchos* (Bleeker)
Fossil record: Late Miocene: Borneo (Brunei Darussalam) [159].

*Carcharhinus albimarginatus* (Rüppell)

Fossil record: Late Miocene–Early Pliocene: Chile (North Coast) [114], Ecuador (Camarones River) [160]; Middle Miocene–Pliocene: Chile (Northern Chile) [42]; Pliocene: Chile (Bahía Inglesa) (Long 1993) [114].

*Carcharhinus amboinensis* (Müller & Henle)
Fossil record: Late Miocene: Borneo (Brunei Darussalam) [159].

*Carcharhinus brachyurus* (Günther)
Remarks and fossil record: This species can be traced back to the Early Miocene [161]. The species were found in a lot of Neogene and Pleistocene localities in Europe, North and South America, Australia, and Japan [161]. It had an Early Miocene East Pacific–Central West Atlantic center of origin [161]. The present-day distributional pattern of *C. brachyurus* is the product of historical biogeographic processes and likely reflects major changes in the global ocean system, including the closure of major seaways and the emergence of new oceanic circulation patterns [161]. Landini et al. [44,161,162] also identified the oldest copper shark nursery area in the East Pisco Basin of Peru, from the Early Miocene of the Chilcatay Formation and the Late Miocene of the Pisco Formation.

*Carcharhinus brevipinna* (Müller & Henle)
Fossil record: Miocene: India (Orissa) [134]; Late Miocene: Panama (Lago Bayano) [129]; Middle Miocene to Early Pliocene: USA (Florida) (as "cf.") [128].

*Carcharhinus falciformis* (Bibron in Müller & Henle)
Fossil record: Early to Late Miocene: Malta [67]; Middle Miocene: India (Kutch) [163], USA (North Carolina) [32]; Middle Miocene to Early Pliocene: USA (Florida) [128]; Late Miocene: Borneo (Brunei Darussalam) [159], Panama (Northern Panama, Lago Bayano) [129,164]; Late Miocene–Early Pliocene: Costa Rica [107]; Pliocene: USA (North Carolina) [155]; Early Pliocene: Italy (Tuscany) [165].

*Carcharhinus glaucus* (Linnaeus) (syn. *Prionace glauca*, see da Silva Rodrigues-Filho et al. [166].
Fossil record: Miocene: Sri Lanka [167]; Middle Miocene–Pliocene: Chile (Northern Chile) [42]; Late Miocene: ?Belgium (Antwerp International Airport) [168]; Late Miocene to Early Pliocene: Chile (Northern Chile) [169]; Early Pliocene: Italy (Parma) [90]; Late Pliocene: Italy (Umbria, Tuscany) [85,170].

*Carcharhinus leucas* (Valenciennes in Müller and Henle)
Fossil record: Early Miocene: Egypt (Moghra) [171], Peru (Zamaca) [44]; Middle Miocene: India (Kutch) [163], USA (North Carolina) [32]; Middle Miocene to Early Pliocene: USA (Florida) [128]; Middle Miocene–Middle Pliocene: Venezuela [151]; Late Miocene: Panama (Northern Panama) [164], Portugal (Alvalade Basin) (as "cf.") [124]; Late Miocene: Peru (Pisco Basin) [172]; Pliocene: Italy (Tuscany) [173], USA (Florida) [154]; Early Pliocene: USA (North Carolina [32]; Canary Islands (Gran Canaria, Fuerteventura) [174], South Africa (Langebaanweg) [175].

*Carcharhinus limbatus* (Müller & Henle)
Fossil record: Miocene: India (Orissa) [134]; Early Miocene: USA (Delaware) (Purdy 1998) [176]; Early Miocene to Late Pliocene: Colombia (Guajira Peninsula) (as "cf.") [95]; Middle Miocene to Early Pliocene: USA (Florida) [128]; Early Pliocene: Italy (Tuscany) [177].

*Carcharhinus longimanus* (Poey)
Fossil record: Early Miocene: India (Kathiawar, Piram Island, Orissa) [178,179]; Pliocene: Italy (Tuscany) [173], Spain (Alicante) [180]. Cappetta [181] identified a tooth from the Pliocene of North Carolina, USA, as *Pterolamiops longimanus*. *Pterolamiops* is a

junior synonym of *Carcharhinus* [182], but, according to Purdy et al. [32], Cappetta's tooth may belong to *C. leucas.*

*Carcharhinus macloti* (Müller and Henle)

Fossil record: Miocene: India (Orissa) [134]; Early Miocene: Brazil (Northeastern Amazonia) (as "cf.") [183], Peru (East Pisco Basin) [73]; Middle Miocene: USA (North Carolina) [32]; Late Miocene: Peru (Cerro Colorado) [184], Portugal (Lisbon) [80].

*Carcharhinus obscurus* (Lesueur)

Fossil record: Early Miocene: Egypt (Moghra) [171]; Mexico (Baja California) (as "cf.") [185]; Venezuela (as "cf.") [186]; Early to Middle Miocene: Cuba [187]; Middle Miocene: Grenada (Carriacou) [188]; Middle to Late Miocene: Ecuador (Carretera Flavio Alfaro) [160]; Middle Miocene–Middle Pliocene: Venezuela [151]; Middle Miocene–Pliocene: Chile (Northern Chile) [42]; Late Miocene: Portugal (Alvalade Basin) (as "cf.") [124], Panama (Northern Panama, Lago Bayano) [129,164]; Pliocene: Italy (Tuscany) [173]; Early Pliocene: USA (North Carolina) [32].

*Carcharhinus perezi* (Poey)

Fossil record: Early Miocene: Brazil (North Brazil) [189], USA (Delaware) [176]; Early to ?Middle Miocene: Venezuela (Falcón Basin) [190]; Early Miocene to Late Pliocene: Colombia (Guajira Peninsula) (as "cf.") [95]; Middle Miocene: USA (North Carolina) [32]; Early to Middle Miocene: Cuba [187]; Late Miocene: Panama (Northern Panama) [164], Portugal (Alvalade Basin) [124]; Pliocene: Italy (Tuscany) [173]; Early Pliocene: USA (North Carolina) [32].

*Carcharhinus plumbeus* (Nardo)

Fossil record: Early Miocene: Italy (Piedmont) [191]; Middle Miocene: USA (North Carolina) [32]; Middle Miocene to Early Pliocene: USA (Florida) [128]; Middle Miocene–Middle Pliocene: Venezuela [151]; Late Miocene: Panama [164], Portugal (Alvalade Basin) (as "cf.") [124]; Pliocene: Italy (Tuscany) [173]; Early Pliocene: USA (North Carolina) [32].

*Carcharhinus sealei* (Pietschmann)

Fossil record: Late Miocene: Borneo (Brunei Darussalam) [159].

*Glyphis glyphis* (Müller & Henle)

Fossil record: Early Miocene to Pliocene: Portugal [192]; Late Miocene: Borneo (Brunei Darussalam) (as "cf.") [159]; Pliocene: Italy (Toscana) [193].

*Negaprion brevirostris* (Poey)

Fossil record: Early Miocene: India (Orissa) [179], Peru (Zamaca) [44]; Early to Middle Miocene: Cuba [187]; Middle to Late Miocene: Ecuador [160]; Middle Miocene–Middle Pliocene: Venezuela [151]; Middle Miocene to Early Pliocene: USA (Florida) [128]; Late Miocene: Panama (Northern Panama, Lago Bayano) [129,164], Peru (Cerro Colorado) [184]; Pliocene: Angola (as "cf.") [156], USA (Florida, North Carolina) [154,155].

Sphyrnidae Gill

*Sphyrna media* (Linnaeus)

Fossil record: Early Miocene: Brazil (Northeastern Amazonia) (as "cf.") [183]; Middle Miocene: USA (North Carolina) (as "cf.") [32]; Late Miocene: Peru (Cerro Colorado) [184]; Pliocene: USA (North Carolina) (as "cf.") [32], Ecuador [160]; Late Pliocene–Pleistocene: Ecuador (Punta Canoa) [160].

*Sphyrna mokarran* (Rüppell)

Fossil record: Early Miocene: Cuba (Domo de Zaza) [194]; Middle Miocene to Early Pliocene: USA (Florida) [128]; Late Miocene: Panama (Lago Alajuela, Northern Panama, Lago Bayano) [129,152,164,195], Borneo (Brunei Darussalam) (as "cf.") [159].

*Sphyrna zygaena* (Linnaeus)

Teeth similar to this species have been found in sediments dating from the Early Miocene and younger [28]. However, there is debate as to whether these teeth belong to *S. zygaena* or to *Sphyrna laevissima* (Cope), described from the Miocene of Maryland, USA [28,32].

## 5. Outlook and Conclusions

Despite a fossil record consisting mostly of teeth, new findngs and methods have increased our knowledge of fossil shark species as well as the fossil records of extant species. In particular, isotopic analyses and computer-based 2D and 3D reconstructions are valuable tools for the study of fossil shark teeth. Paleobiological details surpassing descriptions of the teeth are known for a total of 19 extinct Neogene shark species, with most of the research focused on the well-known, large-bodied *Otodus megalodon*. Aside from the latter taxon, there are no hypotheses developed to date regarding potential causes underlying the extinction of these shark species; however, climate change and habitat loss have been suggested [196]. Concerning the fossil records of the more than 500 extant shark species, 38 species could be verified as present in the Neogene record. Four of these 38 species (*Alopias superciliosus*, *Galeorhinus galeus*, *Rhincodon typus*, and possibly *Alopias vulpinus*, 11%) first appeared during the Paleogene. For five extant species (*Carcharias taurus*, *Hexanchus griseus*, *Isurus oxyrinchus*, *Notorynchus cepedianus*, *Sphyrna zygaena*), the relationship of the extant and fossil forms is not clear. Figures 4 and 5 show the phylogenetic relationships and summarize the stratigraphic ranges of the species discussed in the text. The taxa are divided into Charchariniformes (Figure 5) and non-Carcharhiniformes (Figure 4) for readability. Determining the exact number of shark species present during the Neogene is highly speculative, if not impossible, although it can be assumed that the ancient diversity was similar to the extant diversity, with the addition of now extinct taxa. Reasons for this lack of knowledge include collecting bias (especially concerning minute-sized teeth), incomplete preservation of the teeth, and the poorly known dentition of extant relatives (here, also especially the small species with minute-sized teeth and also the presence or absence of different forms of heterodonty). Sometimes, only one tooth with a different shape is found in a sample, which is not enough for a reliable taxonomic diagnosis, for example, "*Carcharhinus* sp." from Äpfingen, Baden-Württemberg [197].

The implementation of the new methods mentioned herein, extensive collection (especially of minute teeth), and detailed descriptions of the dentition of Recent species will enhance our knowledge of shark evolution and the paleobiology of fossil sharks.

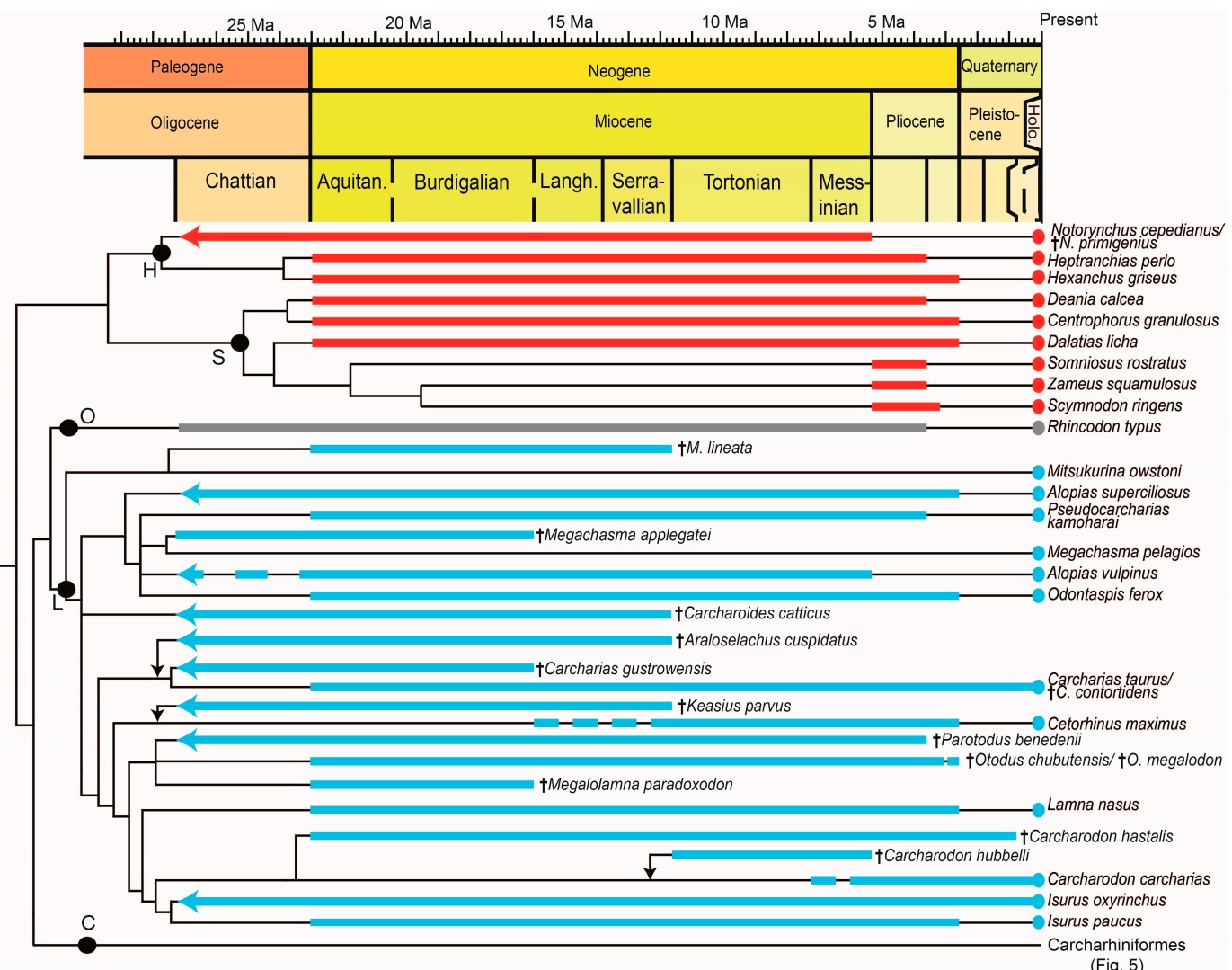

**Figure 4.** Relationships and stratigraphic ranges of non-carcharhiniform species discussed in the text. Topology derived from Stein et al. [198] for extant species, with position of extinct taxa following the review presented here. Branch arrows indicate phylogenetic uncertainty; range arrows indicate taxa that appeared prior to the Late Oligocene; and dashed range lines indicate stratigraphic or taxonomic uncertainty. Node positions not to scale. "Fig. 5" refers to Figure 5. C, Carcharhiniformes; H, Hexanchiformes; L, Lamniformes; O, Orectolobiformes; S, Squaliformes.

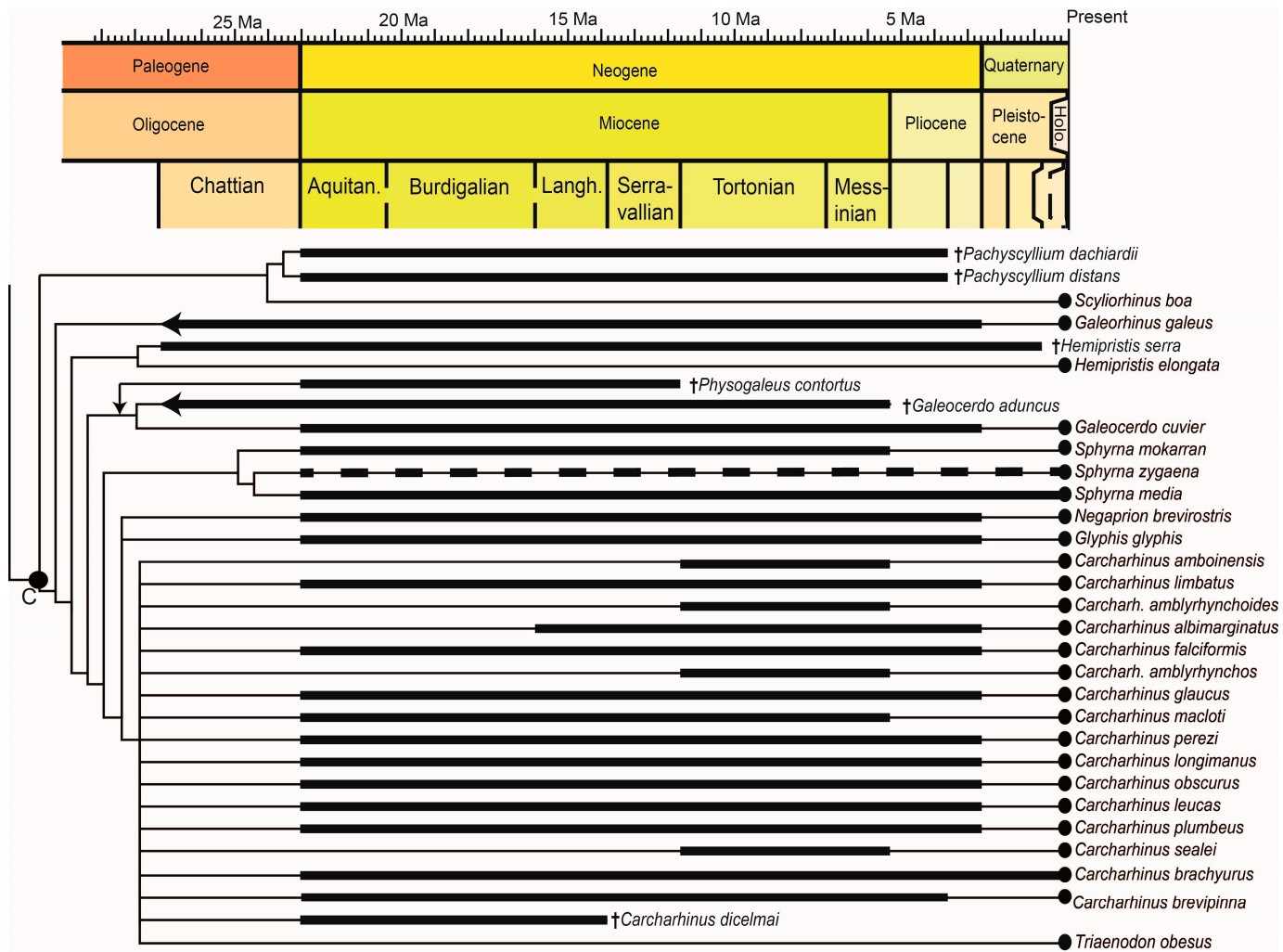

**Figure 5.** Relationships and stratigraphic ranges of carcharhiniform species discussed in the text. Topology derived from Stein et al. [198] for extant species, with position of extinct taxa following the review presented here. Branch arrows indicate phylogenetic uncertainty; range arrows indicate taxa that appeared prior to the Late Oligocene; and dashed range lines indicate stratigraphic or taxonomic uncertainty. Node positions not to scale. C, Carcharhiniformes.

**Author Contributions:** Conceptualization, O.H., E.E.M. and M.W.R.; methodology, O.H., E.E.M. and M.W.R.; investigation, O.H.; resources, O.H., E.E.M. and M.W.R.; data curation, O.H., E.E.M. and M.W.R.; writing—original draft preparation, O.H.; writing—review and editing, E.E.M.; visualization, E.E.M.; supervision, O.H., E.E.M. and M.W.R.; project administration, O.H., E.E.M. and M.W.R. All authors have read and agreed to the published version of the manuscript.

**Funding:** This research received no external funding.

**Institutional Review Board Statement:** Not applicable.

**Data Availability Statement:** No new data were created or analyzed in this study. Data sharing is not applicable to this article.

**Acknowledgments:** We wish to thank Alberto Collareta (University of Pisa, Italy) and Jürgen Pollerspöck (Stephansposching, Bavaria, Germany), as well as Kenshu Shimada (DePaul University, Ilinois, USA), for the permissions to use their photographs of shark teeth. We also thank two anonymous reviewers for their suggested improvements to the manuscript.

**Conflicts of Interest:** The authors declare no conflicts of interest.

## Abbreviations

| | |
|---|---|
| LACM | Natural History Museum of Los Angeles County, Los Angeles, California |
| MUSM | Museo de Historia Natural de la Universidad Nacional Mayor de San Marcos, Jesús María, Lima, Peru |
| SMNS | Staatliches Museum für Naturkunde Stuttgart, Stuttgart, Germany |
| UCMP | University of California at Berkeley, Museum of Paleontology Berkeley, California, USA |

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
