# Peer review of "A Review of the Paleobiology of Some Neogene Sharks and the Fossil Records of Extant Shark Species"

_diversity, doi:10.3390/d16030147_

Round 1

Reviewer 1 Report

Comments and Suggestions for Authors

Dear authors:

The manuscript titlesd “A review of the paleobiology of some Neogene sharks and the fossil record of extant shark species” is very interesting and in my opinion, this review manuscript provides a comprehensive recollection of the fossil record and palaeobiological data from extant Neogene sharks that will be useful for other researchers in the field. The authors have researched the literature extensively, and have incorporated the latest papers showcasing the different methods used to determine the palaeobiological information of this group of organisms, as well as the latest “trends” in the research of this particular field, as shown in the increase of papers focused on stable isotopes.

Despite all the expose above, I have some major comments and /or suggestions that I would like to ask the authors to consider in order to improve the quality of the manuscript:

-First, there are no figures attached to the main manuscript. Furthermore, although the authors had included five figure captions, only two (figure 4 and figure 5) are cited in the main text. This is a major problem that needs to be addressed.

-Second, I would also suggest that the authors add a table in which they can summarize which methods (stable isotopes, bite marks, 2D or 3D computer modelling, etc.) are employed to each of the species they list. The manuscript is dense, so a table to summarize this information would help to better present the information.

-Third, there seems to be some discrepancies in the way the extinct and extant sharks are discussed in the main text. Whereas the authors present the information in a more descriptive way for the extinct sharks, for the extant ones they present the information in a more telegraphic way. I would recommend that the authors follow the same pattern in both parts. So, I suggest that the authors included a fossil record section when discussing the extinct shark species, as they have done with the extant sharks. This would help to homogenize the text and present the information in a more orderly way. Then a remarks and/or a palaebiology section can be added to discus the know aspects of these extinct animals, from the possible position in the phylogeny and ongoing debate about it; to changes in names, etc.

- In page 4, in Odontaspididae, the authors describe Carcharoides catticus, but at the same time, discus C. totuserrataus. Please, separate them as have been done to all other species.

- In page 4, in Carchariidae, the authors describe two different species at the same time: Carcharias contortidens and Carcharias gustrowensis. Please, separate them as have been done to all the other species.

- In page 5, form line 213 to line 222, authors described palaeobiological aspects referred to Carcharias taurus. I would recommend to the authors to move this section to page 13, line 640.

- In page 7, lines 324 to 330, the authors cited some works for the possibility of paleo-nurseries of O. megalodon. I would recommend to the authors to look the work of Herraiz et al. (2020). Use of nursery areas by the extinct megatooth shark Otodus megalodon (Chondrichthyes: Lamniformes). https://doi.org/10.1098/rsbl.2020.0746 and to incorporate it in this section.

-Some references seem to be lacking in the main text or in the references section. Please, see the comments in the PDF attached.

For all the stated above, my recommendation it is to reconsider this manuscript after major revisions.

Author Response

Changes in the formulation were marked yellow in the text

-First, there are no figures attached to the main manuscript. Furthermore, although the authors had included five figure captions, only two (figure 4 and figure 5) are cited in the main text. This is a major problem that needs to be addressed.

Figures were added and cited. There had been a problem with the uploading.

-Second, I would also suggest that the authors add a table in which they can summarize which methods (stable isotopes, bite marks, 2D or 3D computer modelling, etc.) are employed to each of the species they list. The manuscript is dense, so a table to summarize this information would help to better present the information.

A table was added

-Third, there seems to be some discrepancies in the way the extinct and extant sharks are discussed in the main text. Whereas the authors present the information in a more descriptive way for the extinct sharks, for the extant ones they present the information in a more telegraphic way. I would recommend that the authors follow the same pattern in both parts. So, I suggest that the authors included a fossil record section when discussing the extinct shark species, as they have done with the extant sharks. This would help to homogenize the text and present the information in a more orderly way. Then a remarks and/or a palaebiology section can be added to discus the know aspects of these extinct animals, from the possible position in the phylogeny and ongoing debate about it; to changes in names, etc.

A detailed fossil record of the often worldwide distributed extinct species would make the script more as twice as long and is also not the aim of the paper. Both parts of the script also deal with different themes (Palaeobiology of extinct species and fossil record of extant ones). Therefore, the style of discussing it is different.

  • In page 4, in Odontaspididae, the authors describe Carcharoides catticus, but at the same time, discus C. totuserrataus. Please, separate them as have been done to all other species.
  • C. totuserrataus was deteletd
  • In page 4, in Carchariidae, the authors describe two different species at the same time: Carcharias contortidens and Carcharias gustrowensis. Please, separate them as have been done to all the other species.
  • The "Carcharias-complex" (C. taurus wit C. contortidens and C. gustrowensis) were re-ordered
  • In page 5, form line 213 to line 222, authors described palaeobiological aspects referred to Carcharias taurus. I would recommend to the authors to move this section to page 13, line 640.
  • See above
  • In page 7, lines 324 to 330, the authors cited some works for the possibility of paleo-nurseries of O. megalodon. I would recommend to the authors to look the work of Herraiz et al. (2020). Use of nursery areas by the extinct megatooth shark Otodus megalodon (Chondrichthyes: Lamniformes). https://doi.org/10.1098/rsbl.2020.0746 and to incorporate it in this section.
  • Was added

-Some references seem to be lacking in the main text or in the references section. Please, see the comments in the PDF attached.

Was added

Reviewer 2 Report

Comments and Suggestions for Authors

The manuscript presented by Höltke and collaborators present a review of the record of Neogene sharks. The review is comprehensive for some of the species which have been studied more thorughly. The content present cases which will require further research with the methods described. I do not have many comments other than some important aspects required, like the lack of figures in the manuscript file, which are cited in the document. Other than that, and a few other comments (see below), the manuscript would be ok for publication in my opinion.

Comments:

Throughout all of the text the manuscript deals with extinct and extant species, please indicate all extinct species with daggers for clarity.

Abstract

L12 Megaselachus

Introduction

L38 and 39: Erase the paragraph break

L39 Reference change, better:

Andreev PS, Coates MI, Shelton RM, Cooper PR, Smith

PM, Sansom IJ (2015) Upper Ordovician chondrichthyan-

like scales from North America. Palaeontology

58:691–704

L42 Although Benton's reference to list the characters of Neoselachians is useful, it should be acknowledged that a consistent set of traits to unify them (synapomorphies) are controversial, see:

Maisey, J. G. (2012). What is an ‘elasmobranch’? The impact of palaeontology in understanding elasmobranch phylogeny and evolution. Journal of Fish Biology, 80(5), 918-951.

Compagno, L. J. (1977). Phyletic relationships of living sharks and rays. American zoologist, 17(2), 303-322.

and  

Maisey, J. G., Janvier, P., Pradel, A., Denton, J. S. S., Bronson, A., Miller, R., & Burrow, C. J. (2019). Doliodus and pucapampellids: contrasting perspectives on stem chondrichthyan morphology. Evolution and development of fishes, 1, 87-109.

Janvier, P., & Pradel, A. (2015). Elasmobranchs and their extinct relatives: diversity, relationships, and adaptations through time. In Fish Physiology (Vol. 34, pp. 1-17). Academic Press.

L44 "The notochord of is enclosed in" erase of?

L81 "Recent" in small caps

L82 The ordering of the orders and families "the classification"

L273 erase paragraph break

Author Response

Changes in the formulation was yellow marked in the text.

The "Carcharias-complex" (C. taurus wit C. contortidens and C. gustrowensis) were re-ordered

Figures were added. There had been a problem with the uploading.

Abstract

L12 Megaselachus

changed

Introduction

L38 and 39: Erase the paragraph break

changed

L39 Reference change, better:

Andreev PS, Coates MI, Shelton RM, Cooper PR, Smith

PM, Sansom IJ (2015) Upper Ordovician chondrichthyan-

like scales from North America. Palaeontology

58:691–704

changed

L42 Although Benton's reference to list the characters of Neoselachians is useful, it should be acknowledged that a consistent set of traits to unify them (synapomorphies) are controversial, see:

Maisey, J. G. (2012). What is an ‘elasmobranch’? The impact of palaeontology in understanding elasmobranch phylogeny and evolution. Journal of Fish Biology, 80(5), 918-951.

Compagno, L. J. (1977). Phyletic relationships of living sharks and rays. American zoologist, 17(2), 303-322.

and  

Maisey, J. G., Janvier, P., Pradel, A., Denton, J. S. S., Bronson, A., Miller, R., & Burrow, C. J. (2019). Doliodus and pucapampellids: contrasting perspectives on stem chondrichthyan morphology. Evolution and development of fishes, 1, 87-109.

Janvier, P., & Pradel, A. (2015). Elasmobranchs and their extinct relatives: diversity, relationships, and adaptations through time. In Fish Physiology (Vol. 34, pp. 1-17). Academic Press.

added

L44 "The notochord of is enclosed in" erase of?

changed

L81 "Recent" in small caps

To my knowldge (what was told to me), the word "Recent" in the sense of "extant" is capitalized.

L82 The ordering of the orders and families "the classification"

changed

L273 erase paragraph break

changed

Round 2

Reviewer 1 Report

Comments and Suggestions for Authors

The authors have adressed my comments and made the necessary changes, so the quality of the manuscript has improved significantly.  I have attached the manuscript with two last comments, that the authors can implement rather quickly.

Author Response

The changes were done. Thank you for the review.

Best Regards,

Olaf

Reviewer 2 Report

Comments and Suggestions for Authors

I appreciate the effort done by the authors and I have no further comments.

Author Response

Thank vou for the Review.

Best Regards,

Olaf